# Qualitative Analysis of Chair Tasks in Emotion-Focused Therapy Video Sessions

**DOI:** 10.3390/ijerph191912942

**Published:** 2022-10-10

**Authors:** Ghazaleh Bailey, Júlia Halamová, Mária Gablíková

**Affiliations:** Institute of Applied Psychology, Faculty of Social and Economic Sciences, Comenius University in Bratislava, 82105 Bratislava, Slovakia

**Keywords:** emotion-focused therapy, self-compassion, self-criticism, self-protection, consensual qualitative analysis

## Abstract

One of the key elements of Emotion-Focused Therapy (EFT) is decreasing self-criticism as a secondary, maladaptive emotion within two-chair interventions while simultaneously increasing self-compassion and self-protection as primary, adaptive emotions. Though the concepts of self-compassion, self-protection, and self-criticism are highly acknowledged within psychotherapy research, the verbal articulation of these constructs within therapy sessions is underinvestigated. Thus, this qualitative study aims to examine how self-criticism, self-protection, and self-compassion are articulated by clients in EFT video sessions. Consensual qualitative research was used for data analysis performed by two core team members and one auditor. Three similar domains were considered for all three concepts: behavioural, emotional, and cognitive aspects. The number of self-protection statements was the highest among the states. The behavioural aspect was the most dominant domain for self-protection with the major subdomain ‘*I tell you what I need*’. For self-compassion, the cognitive aspect was the most significant domain containing eight subdomains, such as ‘*I see your bad circumstances*’. The most frequent domain for self-criticism was the behavioural aspect consisting of the two subdomains ‘*I point out your wrong behaviours* and *I give you instructions*’. The findings demonstrate the significance of promoting both self-compassion and self-protection to combat self-criticism. More studies of categorising a broader number of cases among various therapy approaches are necessary to develop a more detailed understanding of clients’ verbalisation of self-compassion, self-protection, and self-criticism within therapy.

## 1. Introduction

### 1.1. Emotion-Focused Therapy

Emotion-Focused Therapy (EFT) operates under the principle that negative emotions can be transformed through positive emotions [1,2]. EFT therapists guide clients through specific tasks developed to transform primary maladaptive emotions, such as shame, into more adaptive emotions, such as self-compassion and self-protection [3,4,5]. EFT exploits two-chair enactments [1], such as the self-critical split (also known as two-chair dialogue) and unfinished business (also known as the empty-chair work), to support clients in expressing and deepening unrecognised emotions and parts of the self in order to come to a resolution [1,6]. The self-critical split is a conversation between two aspects of the self: the critical voice and the criticised self. The key marker is an inner critical voice [6,7,8]. The empty-chair work is a dialogue with an imaginary other [3,9]. The marker is an unresolved, problematic emotional experience with a significant other [9]. Both two-chair techniques intend to reduce self-criticism, shame, and helplessness while enhancing self-compassion and self-protection [1,3]. Within the self-critical split, the EFT therapist encourages the client to express the anger, hate, contempt, or disgust of the critical voice towards the criticised self. Afterwards, the client is asked to change sides and respond as the criticised part of the self to the critic [7,8]. During the empty-chair dialogue, the client is encouraged to first express secondary emotions, such as rejecting anger, and then the client is guided to articulate more primary emotions, such as unmet needs and grief, about the losses and/or compassion for the wounded inner child [9]. The therapist guides the client empathically through these emotional processes encouraging the client to be aware, feel, and express their needs and feelings [5]. Through the expression of self-protection and self-compassion, clients transform problematic emotional reactions into more primary adaptive emotions [4,10,11].

### 1.2. Research on Self-Compassion, Self-Protection, and Self-Criticism

Given the fact that two-chair techniques are essential in EFT [1] and considering the importance of self-talk in psychotherapy research and practice [7,12,13], there is a lack of research on how clients characteristically articulate self-criticism, self-compassion, and self-protection during these interventions. In regards to EFT two-chair tasks, there is one study [14] that looks specifically at the client–therapist interaction during a two-chair self-soothing dialogue by using a conversational analysis and identifying the self-soothing structure. The findings characterize the compassionate voice as caring, positive, and supportive, and focusing on resources and positive qualities [14]. Another study examines the efficacy of the self-critical two-chair dialogue in EFT [8]. The work of Shahar et al. [8] supports the two-chair dialogue as a promising intervention on decreasing self-criticism and increasing self-compassion. However, none of the studies focuses on the utterances of being self-critical, self-protective, or self-compassionate in EFT. Moreover, there is no research on the efficiency of the two-chair dialogue on developing self-protection and the kind of statements clients express in therapy sessions. When expressing self-protection, clients are encouraged to express their needs towards the critic. For example, this could sound like ‘*I need you to understand and support me. I need you to stop putting me down like this*’ [11]. Outside of EFT, there a is recent qualitative study by Halamová et al. [15] detecting the statements people express while talking to themselves self-critically in a two-chair dialogue. Participants were expressing feelings of inadequacy, disgust, contempt, helplessness, pity, disappointment, self-hate, and self-directed anger. On the cognitive spectrum, the statements included negative judgement about one’s self, attributing negative qualities and negative characteristics of one’s self, negative evaluation of one’s self and others, and evaluation of the situation they criticised themselves for. The behavioural utterances involved criticism on how participants behaved towards loved ones and pointing out shortcomings, mistakes, inadequate actions, and disappointing behaviour. Whelton and Henkelman [16] also analysed video recordings of people criticising themselves in a two-chair setting. They evolved the utterances in eight categories: “demands and orders; exhorting and preaching; explanations and excuses; inducing fear and anxiety; concern, protection, and support; description; explore/puzzle/existential; and self-attack and condemnation” ([16], p. 90). Both studies validate previous research characterising self-criticism [16,17,18,19]. In another qualitative study, Halamová et al. [20] categorised the first three associations participants relate to self-criticism and criticism. They grouped the associations for self-criticism into three domains: emotional, behavioural, and cognitive. Their results identified the behavioural aspect as the most significant domain, while the emotional aspect was the least represented one. The behavioural aspect further includes three subdomains: motivational function (e.g., improvement and new beginnings); behavioural expressions (e.g., lecturing); and how to handle criticism (e.g., discipline and understanding) ([20], p. 370). Their findings show that self-criticism is mostly linked to how people behave contrary to their feelings. According to Shahar [18], self-criticism is “an expression of hostility and derogation toward the self” ([18], p. 5) when high standards are not met. In line with this, scholars agree that self-directed criticism is marked by negative self-evaluation, self-judgement, self-blame, perfectionism [21], emphasis on mistakes [22], and self-attack [7,16]. Similarly, Halamová et al. [23] conducted a qualitative study on the free associations of compassion and self-compassion. With regards to self-compassion, they classified the associations into four domains: emotional, behavioural, cognitive, and evaluative. In contrast to self-criticism, the emotional aspect was detected as the most frequent domain. Thus, this reveals that self-compassion is mainly connected to feelings rather than behaviour. Participants linked self-compassion to the positive emotions of love and calmness and the negative emotions to unhappiness, sadness, and remorse. In addition to this, there is a recent study [24] analysing the utterances of people articulating self-protection and self-compassion in two-chair work. The authors compared self-protective and self-compassionate statements of participants responding to their inner critical voice. Like the other qualitative studies, the domains included emotional, cognitive, behavioural, and an additional domain, interpersonal aspects. The study shows that self-compassionate statements include, e.g., showing understanding for the situation (cognitive), motivation to activate behaviour in difficult times (behavioural aspect), and the expression of positive emotions towards self (emotional aspect). Their results are in agreement with previous research characterising self-compassion [25,26]. Strauss et al. ([25], p. 19) determine self-compassion as a multidimensional construct consisting of cognitive, affective, and behavioural aspects involving the following five components:“recognizing suffering;understanding the universality of suffering in human experience;feeling empathy for the person suffering and connecting with distress (emotional resonance);tolerating uncomfortable feelings aroused in response to the suffering person (e.g., distress, anger, fear) so remaining open to and accepting of the person suffering; andmotivation to act/acting to alleviate suffering.”

Germer and Neff [26] define self-compassion as compassion for one’s self, which comes along with positive feelings of warmth and care for one’s self. Thus, along with self-criticism, self-compassion is recognised as a multidimensional concept including emotional, behavioural, and cognitive elements. To the best of one’s knowledge, this is the only study on the verbal expression of self-compassion so far. In regards to self-protection, there has been a growing interest in the construct of self-protection as an important element to alleviate self-criticism [11,27,28]. As previously mentioned, there has only been one study on the verbal expression of self-protection in two-chair work [24]. The outcome of the study indicates that peoples’ self-protective responses towards their inner critic emphasise the right to behave and stand up for one’s self (behavioural aspect), the right to decide on their own (cognitive aspect), and the right to feel (emotional). In EFT, self-protection is represented as the ability to express unmet needs in an assertive manner at maltreatment [4,11]. Moreover, it is defined by “a sense of entitlement to be loved, acknowledged, and secure” ([11], p. 35). By expressing self-protection, clients develop a sense of agency and increased strength to stand up for their own needs [11,27,28]. Hence, self-protection is marked as energetic, strong, empowered, resilient, and firm [11].

Over recent years, more studies have been shedding light on the effectiveness of EFT on self-compassion, self-criticism, and self-protection. Halamová et al. [27] investigated the efficiency of an emotion-focused training session on self-compassion and self-protection and the psychological and physiological effects of the training [29]. Additionally, she designed an emotion-focused training session for emotion coaching [30] and examined the effectiveness of the training on reducing self-criticism. Ample research has been drawing attention to measurement instruments for self-criticism [17] and self-compassion [25,31,32] and to the relationship between these two states [13,22,28,33]. Despite the fact that the significance of inner self-talk is widely known and acknowledged in literature [12,13,34], it is an underinvestigated area in psychotherapy research. Though everyone has a self-directed inner critical voice to a certain degree, self-criticism in psychotherapy is connected with a number of psychological disorders, such as depression [21,35], social anxiety [36], and eating disorders [37]. In contrast, self-compassion correlates with emotional balance, less anxiety, shame, and depression; and better mental and physical health [1,38,39,40]. Regarding the effect of an inner self-protective voice, research so far has focused on the efficiency of assertiveness skills training on mental health [41,42,43]. However, in EFT, both self-protection and self-compassion are recognised as inevitable to combat self-criticism [4,11].

## 2. The Aim of This Research Study

To date, there are currently no studies that empirically identify how clients articulate self-criticism, self-compassion, and self-protection in EFT sessions. We want to provide practitioners with an inner atlas of how these states are verbalised by clients during chair work. Hence, having an inner road map can support therapists in facilitating the expression of each state in a productive manner and make the process even more efficient. This way, clients can experience these constructs in an efficient way, which is crucial for promoting therapeutic change [44]. Therefore, the current qualitative study explores the following research question: How are self-criticism, self-protection, and self-compassion verbally expressed by clients within an EFT session during two-chair work?

## 3. Materials and Methods

Due to ethical limitations and the requirements needed for the research, we decided to examine commercially available videos. The APA, CPCAB, and P&E Films offer tapes of real EFT sessions that can be used for research. The video tapes needed the following requirements: they needed to be in English;needed to involve sequences of clients expressing self-compassion, self-protection, or self-criticism within the therapy session in two-chair work;the quality of the tapes had to be sufficient for the transcription.

A total of seventeen EFT sessions were reviewed by the first author and consulted with the second author. As a result, twelve met the criteria. Four sessions incorporated two-chair dialogues with the critical self, and eight illustrated empty-chair dialogues with a significant other. The therapy sessions were led by leading EFT experts. All the clients were female. The EFT videos used in this study address the following topics:

Leslie Greenberg as therapist: Sessions 2, 3, 4, and 6 from EFT over time with Marcy. Psychotherapy in six sessions [45]: EFT for Depression with Dione [46]; Working with core emotion with Darum [47]; Working with current and historical trauma with Sam [48]; Working with social anxiety with Dawn [49]; with Sandra Paivio as therapist: Narrative processes in EFT with Hannah [50]; with Rhonda Goldman as therapist: Case formulation in EFT. Addressing unfinished business with Candy [51]; and with Ladislav Timulak as therapist: Transforming Emotional Pain: An Illustration of Emotion-Focused Therapy video with Claire [52].

### 3.1. Procedures and Data Analysis

As EFT therapists, the first and second authors had the qualifications to select sequences segmenting self-compassion, self-protection, and self-criticism in the therapy videos. Sections were determined based on the consensus of the first and second authors. Each state was selected according to EFT criteria for productive emotion processing mentioned in Auszra’s productivity model (Auszra and Greenberg, 2008), Pascual-Leone’s CAMS (Pascual-Leone and Greenberg 2005), and Hermann’s Emotion Category Coding System [53].

Self-protection: Strong, firm voice and body posture, expression of needs, wishes, and wants. Feeling entitled to have feelings and needs. Stopping maltreatment and protecting one’s self [11].

Self-compassion: Kind, caring, softer voice, acknowledging pain, showing love, understanding, care, giving compassionate suggestions on how to alleviate suffering, and feeling sorry for maltreatment. Empathising with the suffering, showing understanding for the suffering or distress, and showing dedication to alleviate suffering (Strauss et al., 2016).

Self-criticism: Harsh voice. Expression of contempt, anger, disgust, shame, rejection, criticism, negative self-evaluation, self-hate, and self-judgement (e.g., Pascual-Leone et al., 2013).

The audios were transcribed by the first author with the help of the acoustic program Praat [54]. Ultimately, 8 transcripts for self-compassion, 7 transcripts for self-criticism, and 10 transcripts for self-protection were created. Altogether, 96 statements for self-compassion, 109 statements for self-criticism, and 170 statements for self-protection were collected. Consensual qualitative research [CQR; 55] was used for the analyses as CQR is designed to investigate precisely certain aspects of the experience. Furthermore, the method helps to avoid bias.

The transcribed utterances were defined for each state by the first author. Concerning self-criticism, we coded all statements in the transcripts that were expressed by the critical voice in the two-chair dialogue. Attention was paid to phrases characteristic for self-criticism, such as pointing out mistakes, giving instructions, expressing negative self-evaluation [18], contempt, disgust, and anger [7,16]. In terms of self-compassion, all utterances verbalising compassion on both sides (experiential self and critical voice) of the chair work were encoded. The data included kind and caring statements and a motivation to alleviate suffering, such as defined by Strauss et al. [25]. For self-protection, all data in the transcript in which the clients were showing self-protection towards the critic were decoded, standing up for themselves, setting boundaries, and stopping maltreatment. The material contained self-protection attributes, such as expressing needs and standing up for one’s self in an assertive manner [11].

### 3.2. Research Team and Consensual Qualitative Research

The CQR team consisted of two researchers (GB and MB) and one auditor (JH). All are trained in EFT approaches and work in psychotherapy and counselling settings. CQR involves the individual development of the domains, subdomains, and categories by all team members. A main advantage is that the team and auditor must achieve consensus [55]. Our team agreed on the three domains from a previous pilot study [56] before each team member started developing separate subdomains, categories, and characteristics. After the first round, the team came together and discussed the core ideas. The results were sent to the auditor who gave feedback. Afterwards, the team came together in a second round and discussed further results until they reached a consensus. After the auditor’s second feedback and a final group discussion, the changes were integrated. The final results were confirmed by the auditor.

## 4. Results

The outcome of the consensual qualitative analysis is as follows.

For the examples see Table 1, Table 2 and Table 3.

### 4.1. Self-Criticism

From the total coded statements for self-criticism (*n* = 109), the consensus between the coders and the auditor revealed 8 subdomains, 19 categories, and 34 characteristics. The most frequented domain was the behavioural aspect (f = 50, 46%). The second domain was the cognitive aspect (f = 42, 38%), and the least frequented domain was the emotional aspect (f = 17, 16%). The emotional aspect included the processing and identification of emotional experiences of the participants, while the cognitive aspect represented mainly thoughts, self-evaluation, and evaluation of criticised situation and the behavioural aspect described specific behaviours towards one’s self and others.

#### 4.1.1. Behavioural Aspect

The behavioural aspect for self-criticism was the most represented domain. This domain represented all the statements focusing on past, present, and future behaviour. It consists of the two subdomains ‘*I point out your wrong behaviour*’ (contained all data concentrating on disapproved behaviour by the critic) and ‘*I tell you what to do*’ (covered all critical sentences telling the experiential self what to do instead and what not to do). In the first subdomain and in the first category ‘*I put you down for past behaviours*’, all statements by the critic saying which behaviour in the past was wrong or not enough were included. The category ‘*I doubt your present behaviour*’ questioned current actions, while ‘*I mistrust your future actions*’ contained utterances questioning future actions. The second subdomain ‘*I tell you what to do*’ was divided into the two categories: ‘*What to do instead*’ (consisted of instructions on what to do more) and ‘*What not to do*’ (orders saying which behaviour should be stopped).

#### 4.1.2. Cognitive Aspect

The cognitive aspect was the second recurrent domain with four subdomains. It displayed all data involving negative self-evaluation by the critic. The first subdomain was termed ‘*I evaluate you as a person*’. This subdomain included statements involving the critic expressing high standards when standards are not met. Beginning with ‘*I have high standards towards you*’, this category represented statements characteristic for not meeting standards and setting conditions. The second category was named ‘*You are not perfect*’ and pointed out the utterances by the critic on points of imperfection. The third category ‘*You are inadequate*’ involved sentences illustrating inadequacy. The last category of this subdomain was ‘*You are ridiculous*’ and involved the critic mocking themself. The second subdomain ‘*I evaluate your situation*’ consisted of one category and covered all phrases saying what was wrong with one’s self being in the current situation. The third subdomain was called ‘*You should be*’ and represented statements telling which characteristics of one’s self should be more and which less. Thus, this subdomain was divided into the categories ‘More and Less’. The final subdomain consisted of the critic saying ‘*You don*’*t deserve having needs*’ and contained data rejecting needs.

#### 4.1.3. Emotional Aspect

The least frequent domain of self-criticism was the emotional aspect. This domain covered all negative emotions expressed by the critic towards one’s self and criticisms of emotions felt by one’s self. What you feel is disproportional contained statements determining which feeling is too much and which one is too little. Thus, the categories involved those specific emotions. The second subdomain was named ‘*You are unpleasant to me*’. It involved phrases saying how the critic felt towards the self/client. The categories named specific emotions, such as contempt, shame, anger, and disgust.

### 4.2. Self-Compassion

For self-compassion (*n* = 96), the most frequented domain was the cognitive aspect (f = 51; 53%) followed by the behavioural aspect (f = 31; 32%). The least frequented domain was the emotional aspect (f = 14; 15%). A total of 11 subdomains, 23 categories, and 24 characteristics were coded for self-compassion.

#### 4.2.1. Cognitive Aspect

In terms of self-compassion, the cognitive aspect was the most frequent and also the most saturated, consisting of 8 subdomains, 16 categories, and 13 characteristics. This domain consisted of all the content showing understanding and support. The subdomain ‘*I see your bad circumstances*’ included all statements recognising the difficulty of the situation the client was in. Thus, the categories involved statements justifying the difficult circumstances and their impact on one’s self: ‘*You were a victim*’*,* ‘*It was your childhood*’*,* ‘*It*’*s situational*’. The next subdomain was called ‘*You deserve all universal human needs*’ and approving the clients’ human needs. Hence, the categories included all phrases accepting the need for love and understanding: it is normal to want to be loved and understood. The third subdomain was called ‘*It*’*s ok to have difficult feelings*’. This subdomain contained sentences affirming negative feelings. The categories represented specific feelings, such as anger and frustration: it’s ok to be angry, and it’s ok to be frustrated. ‘*I missed closeness with you*’ reflects the longing for closeness and connection and has no further categories, while ‘*I*’*m grateful for your support*’ summarised all the statements acknowledging the positive aspects of the critic. The categories in this subdomain realised the learnings from the critic and are divided into ‘*You taught me taking action*’, ‘*You helped me feeling empowered*’*;* and ‘*You helped me feeling strong*’. This is followed by the subdomain ‘*I appreciate you,*’ which considered all the data reflecting on personal resources and value. Therefore, the categories indicated the particular appreciations, such as ‘*I appreciate myself for my positive qualities*’, ‘*I appreciate myself for my achievements*’ and ‘*You matter*’. Thereafter comes the subdomain ‘*I accept myself for who I am*’ characterised through all statements expressing self-acceptance within one category called ‘*I*’*m ok*’. The last subdomain involved all compassionate advice from the compassionate voice. Therefore, we named this subdomain ‘*I give you compassionate advice*’. The categories ‘*I suggest you to do*’ and ‘*I suggest what you shouldn*’*t do*’ included instructions regarding self-care, self-confidence, and kindness towards one’s self and others.

#### 4.2.2. Behavioural Aspect

The behavioural aspect identified all behaviour-related statements describing ways the compassionate voice wants to alleviate suffering. Hence, the subdomain is named ‘*I*’*m going to alleviate your suffering*’. Hereto, phrases were considered about which negative, harmful behaviours the critic wants to stop and how they want to be kind and caring. Consequently, the categories ‘*I*’*m going to stop hurting you*’ and ‘*I will show support*’ were termed.

#### 4.2.3. Emotional Aspect

The emotional domain captured all the statements by the self-compassionate voice expressing emotions towards the self. The first subdomain represented phrases verbalising empathy and was called ‘*I feel your pain*’. The statements for the first subdomain were distinguished in categories empathising with bad feelings and expressing apology: ‘*You felt bad*’, ‘*It*’*s hard*’, and ‘*I*’*m sorry for*’. The second subdomain consisted of data involving the expression of positive feelings. Thus, it was named ‘*I have positive feelings for you*’. The categories described the specific feelings love and pride.

### 4.3. Self-Protection

The data for self-protection (*n* = 167) included 10 subdomains, 26 categories, and 33 characteristics. The most frequented domain in the data was the behavioural aspect (f = 87; 52%). The second was the cognitive aspect (f = 55; 33%), and the third was the emotional aspect (f = 25; 15%).

#### 4.3.1. Behavioural Aspect

In regards to self-protection, the most prevalent domain was the behavioural aspect. This domain implied material expressing behavioural needs towards the critic. The subdomain ‘*I tell you what I need*’ signals a variety of needs towards the critic and towards the self. It goes from a need for acceptance and kindness to a need for having boundaries. Furthermore, this subdomain included all phrases expressing assertion towards the critic. The nine categories described the particular needs and were named ‘*I need you to see me for who I am*’, ‘*I need you to stop negative behaviour to me*’, ‘*I need you to do good to me*’, ‘*I don*’*t need you at all*’, ‘*I need to stop negative behaviour towards myself*’, ‘*I want to take responsibility for my life*’, ‘*I need to be good to myself*’, ‘*I need to set boundaries*’, and ‘*I reject your criticism*’.

#### 4.3.2. Cognitive Aspect

The second most frequent domain of self-protection was the cognitive aspect. Predominately, this domain contained self-protective statements of clients standing up for themselves towards the critic. Starting with the subdomain ‘*I show understanding towards myself*’, this subdomain included statements of clients defending their worth and rights. The related categories described the entitlement to be one’s self: ‘*I have a right to be myself*’; the entitlement to be loved: ‘*I deserve to be loved*’; and the assertion: ‘*You don*’*t have the right to judge me*’. The next subdomain involved phrases recognising the motive of the critic. Therefore, it was termed: ‘*I acknowledge your protective function*’. The categories separated two kinds of acknowledgements: ‘*I acknowledge your effort*’ and ‘*I acknowledge your reasons*’. The last subdomain of the cognitive aspect was called ‘*I criticize you back*’. We included all the material saying what the critic is not doing or doing wrong. It was the most saturated one consisting of six categories differentiating the different mistakes of the critic: ‘*You don*’*t acknowledge me*’, ‘*You*’*re forcing me*’, ‘*You don*’*t listen*’, ‘*You always change your standards*’, ‘*You don*’*t care about me*’*, and* ‘*You*’*re not nice to me*’.

#### 4.3.3. Emotional Aspect

The least frequent domain of the self-protective voice was the emotional aspect. It involved all data articulating emotions towards the critic and towards the self. This domain was separated into six subdomains. The subdomain ‘*I don*’*t want you to make me feel*’ consisted of client statements saying which negative feelings they do not want to have anymore. The feelings were categorised as: frightened, lonely, worried, weak, and tired. The next subdomain involved positive feelings towards the self and was named ‘*I*’*m proud of myself*’. Another subdomain consisted of statements expressing anger towards the critic. Hence, it was termed ‘*I*’*m angry at you*’, which was divided into six categories justifying the anger: ‘*Rejecting my feelings*’, ‘*Not being there for me*’, ‘*Forcing me*’, ‘*Being so negative*’, ‘*Being dismissive*’, and ‘*For living your life through me*’. The last three subdomains represented all sentences naming specific negative feelings, such as: ‘*I*’*m hurt*’, ‘*I*’*m disappointed*’, and ‘*I feel helpless*’.

## 5. Discussion

The goal of the current study was to explore how self-criticism, self-compassion, and self-protection are articulated by clients within EFT sessions during two-chair work. As previously acknowledged, self-compassion and self-protection are both substantial constructs in decreasing self-criticism in EFT [5,11]. Thus, we wanted to enhance the understanding of these three states of the client’s self and how these are communicated in real EFT sessions. By applying consensual qualitative research, the three domains: emotional aspect, cognitive aspect, and behavioural aspect arose for all three constructs. The most dominant domain for both self-criticism and self-protection was the behavioural aspect, the second the cognitive, and the third the emotional aspect. This signifies that clients expressing self-criticism and self-protection mainly focus on behaviours and less on emotions and therefore must be facilitated to focus on better outcomes by their therapists [44]. For self-compassion, the cognitive aspect was highlighted as the most frequent domain, followed by the behavioural aspect. Interestingly, the emotional aspect was the least dominant domain as well. Thus, clients were focusing mostly on cognitively understanding the bad circumstances, deserving human needs, accepting difficult feelings, expressing gratefulness, acceptance, appreciation, and giving self-compassionate advice rather than on behaviours and emotions.

### 5.1. Behavioural Aspect

#### 5.1.1. Self-Criticism

The behavioural aspect was the most frequent domain for self-criticism. The self-critical voice in the therapy sessions was very much action-oriented. Clients’ inner critics were focusing on pointing out past and present wrong behaviours, doubting future actions, and telling what to do instead and what not to do. This goes along with another qualitative analysis on the verbal expression of self-criticism by Halamová et al. [15]. In their study, participants were, among others, criticising themselves for doing something wrong in their lives. It is noteworthy that they were not telling themselves what to do. In alignment with this, it is generally known that indicating mistakes is one of the most common aspects of self-criticism [18,19,57]. According to G. Shahar [18], self-criticism is characterised by the expectation of unreachable high standards and the articulation of hatred and degradation when those standards are not achieved. In our study, clients were putting themselves down for not doing things right or enough and expressing doubt and mistrust in present and future actions. Doubting one’s own decisions and behaviours and being ambivalent about one’s self is one of the key components of depression [58]. Having said that, ample research has acknowledged the link between self-criticism and depression [57,58,59]. In EFT, the self-critical two-chair dialogue is one of the main interventions in the treatment of depression [59,60]. It is important to mention that two of our analysed videos focused on EFT for depression [45,46]. Thus, our results provide support that self-criticism is related to depression. As previously mentioned, clients were not only pointing out wrong behaviours. In addition, the self-critic was giving instructions on what they should do rather than what they should not do. The inner critical voices were pushing clients to do more (more work, more money) and obey the critic. Our findings are in line with Whelton and Henkelman’s’ [16] findings in which the category demands and orders emerged out of the self-critical statements. Self-critical people are very much focused on avoiding mistakes and failures [18,57,61]. One way they try to reduce the possibility of failure is through control [57]. Thus, it is assumed that giving instructions is a way for self-critical people try to control their behaviour in order to limit the probability of unsuccessfulness. This includes rejecting feelings (clients were saying don’t feel, don’t cry). By telling themselves not to feel or cry, self-critical people try to avoid being abandoned and consequently feeling lonely. Therefore, our results add to the understanding that depression is associated with the fear of abandonment and loneliness [57,59]. Lastly, our results are consistent with Halamová et al. [20]. In their qualitative study, the authors determined the behavioural aspect as well as the most frequent domain associated with self-criticism.

#### 5.1.2. Self-Protection

With the greatest number of statements, the behavioural aspect of self-protection is the most saturated domain. Overall, of all three states, self-protection is the state verbalised by the analysed cases. This supports the essentiality of self-protection as a counteract to self-criticism [5,62] as opposed just to prioritising self-compassion [38]. In our study, this domain included statements of clients telling the critic what they need (I tell you what I need). They verbalised two different forms of needs. On the one side, they asked the critic to stop negative behaviour and be good. On the other side, they voiced to the critic what they need to do for themselves. These utterances included assertive needs, such as I need to stop negative behaviour towards myself, I want to take responsibility for my life, I need to be good to myself, and I need to set boundaries. This is in line with a qualitative study by Koróniová et al. [63]. They discovered that low self-critical participants indicated the need to stand up towards and to stop the critic. A defining aspect of EFT is helping clients to access and transform core painful experiences by bringing them into awareness and encouraging them to express their underlying unmet needs [4,64]. Hence, the ability to assert and support one’s own needs is a required quality of self-protection [4,11] and a healthy way to increase self-criticism. Correspondingly, Pascual-Leone and Greenberg [4] recognize setting boundaries as a significant goal of self-protection. Furthermore, Alike and Sedikides [65] determine self-protection as a motivation for people protecting themselves towards negative self-views. Moreover, the authors describe self-protection as a mechanism that comes up when individual interests are threatened. In our cases, clients felt empowered to stand up for themselves by letting the critic know that they do not have the right to judge and that they have the right to be themselves. Similarly, Vráblová et al. [24] investigated in their consensual qualitative research a behavioural domain in which their participants expressed limits, argued for their rights and stood up for themselves in an assertive manner. Thereupon, our results confirm previous research on the importance of expressing needs towards maltreatment in order to feel stronger and more resilient [5,11].

#### 5.1.3. Self-Compassion

As the second most recurring domain, the behavioural aspect of self-compassion contained the subdomain, ‘*I*’*m going to alleviate your suffering*’. In our cases, the self-compassionate voice showed motivation to alleviate the suffering by showing support (I will show support) and by stopping hurtful behaviour (I’m going to stop hurting you). There is a broad consensus that the desire to decrease suffering is a significant aspect of compassion [25,40,66]. Strauss et al. [25] define this fifth element of compassion as a “motivation to act/acting to alleviate suffering” ([25], p. 19). Goetz et al. [66] identify compassion as a feeling that emerges with the desire to help when seeing the suffering of others. Accordingly, Pauley and McPherson [67] indicated in their study that people generally understand and experience compassion through compassionate behaviours. Likewise, Halamova et al. [23] discovered in their qualitative analysis on free associations of compassion and self-compassion that participants were associating self-compassion with displaying self-support. In our study, the self-compassionate part was showing support by offering a safe place, comfort, and something uplifting. Furthermore, clients were expressing confirmation by telling themselves ‘*You did your best*’ and ‘*It*’*s not your fault*’. This supports the general understanding of self-compassion as an antidote to self-blame and self-criticism [31,68]. Thus, by saying ‘*It*’*s not your fault*’ or ‘*You did your best*’, clients respond with kindness and understanding instead of becoming self-critical in situations of failure. In EFT theory, self-compassion is recognised as a primary adaptive helpful emotion that has a soothing quality [4,62].

### 5.2. Cognitive Aspect

#### 5.2.1. Self-Criticism

The second most dominant domain, the cognitive aspect, involved a negative evaluation of the self, of the situation, and the rejection of needs. Negative self-evaluation is widely acknowledged as a key characteristic of self-criticism [21]. In our study, clients were evaluating themselves by having high standards, expressing imperfection, inadequacy, and ridicule. Clients were verbalising statements making themselves feel generally not enough (e.g., You are not enough as a person, not enough skilled, etc.). Our findings are in line with ample research identifying the establishment of high standards as a significant aspect of self-criticism [18,57]. Furthermore, this domain corresponds with the recent consensual qualitative study by Halamova et al. [15]. Their data support our results that the self-critical voice highlights insufficiencies. Participants in their research were criticising themselves for, e.g., lack of skills and performance. In line with this, another qualitative research by Halamova et al. [15] recognised evaluation as a subdomain of the cognitive aspect. According to Thompson and Zuroff [61], there is a form of self-criticism that emerges in situations when people do not meet internal personal standards. Gilbert et al. [22] call this style of self-criticism inadequate self as it focuses on personal inadequacies. Along with this, Koróniová et al. [63] showed in their study that people criticise themselves particularly through accusations of not meeting expectations. As stated by G. Shahar [18], maladaptive perfectionism also named “excessive-evaluative-concerns” ([18], p. 34) is in fact self-criticism. In our study, these kinds of self-critical statements were characterised as ‘*You*’*re not the best*’ and ‘*You are good, but*’. Moreover, clients were telling themselves what they should do more or less in order to build, e.g., self-esteem. According to Gilbert et al. [22] , self-criticism can also be a self-effort for improvement. However, Powers et al. [69] compared self-orientated perfectionism and self-criticism and found out that self-criticism is notably negatively linked to goal process. In addition, clients in our research were rejecting their own needs, such as the need to be loved. In EFT theory, self-criticism is a form of rejecting anger that covers actual primary maladaptive feelings, such as shame [11]. Hence, it is understandable that self-critical people do not feel entitled to have needs.

#### 5.2.2. Self-Protection

The cognitive aspect was the second most frequent domain for self-protection. In the two chair-dialogue, clients showed understanding towards one’s self, acknowledged the protective function of the critic, and criticised the critic back. The most frequently mentioned subdomain was ‘*I criticize you back*’. In this subdomain, clients were criticising the critic for not listening, not acknowledging them, for changing their standards, and not being caring and nice. Standing up towards the critic by criticising them back and telling them that they do not have the right judge increases a sense of empowerment, which is the goal of self-protection [11]. When clients feel more resilient, they feel entitled to have needs [1,11]. Our clients were arguing to the critic that they have the right to be themselves. This is in accordance with Vráblová et al. [24]. In their qualitative study, the categories ‘*You have the right to say no*’, ‘*You have the right to set your limits*’, and ‘*You have the right to decide*’ situationally emerged. Moreover, our clients were articulating that they deserve to be loved. Self-criticism is acknowledged as a consequence of internalised values [61,70,71] often by someone who has been developmentally significant. Hence, emotional neglect and shaming lead to a sense of unworthiness [71,72]. As clients start feeling stronger and more resilient, they develop self-acceptance and feel entitled to be loved [11,71]. Interestingly, only one client in our study expressed the entitlement to be loved. We suppose this is due to the setting in which the therapy sessions were recorded. The majority of the therapy sessions we analysed were single sessions. As a consequence, clients were seeing the therapists only for one session. According to Pascual-Leone’s and Greenberg’s sequential model of emotion processing [4], self-protection is the third stage of EFT. Thereby, it is reasonable that articulating the right to be loved is something very vulnerable and potentially too soon to express in the first session. Another subdomain was the acknowledgement of the protective function of the critic. Early childhood experiences of criticism, bullying, and neglect evoke shame and self-criticism [72]. Furthermore, negative self-treatment becomes a way to cope with the underlying painful emotions [4,11,71]. In other words, the self-critical voice has developed as a protection mechanism from being hurt again. Thus, acknowledging the protective function of the critic promotes understanding towards the inner critical voice. This helps clients to build self-compassion and self-empathy [71,73].

#### 5.2.3. Self-Compassion

Consisting of eight subdomains, the most comprehensive domain for self-compassion was the cognitive aspect. The self-compassionate voices of our cases acknowledged universal human needs (You deserve all universal human needs), the bad circumstances (I see your bad circumstances), and difficult feelings (It’s ok to have difficult feelings). This goes along with the first two aspects of Strauss et al.’s definition (2016) of self-compassion: “1. Recognizing suffering; 2. Understanding the universality of suffering in human experience” ([25], p. 19). In line with this, Neff (2003b) identifies being kind and understanding towards one’s self in times of pain and failure as a significant element of self-compassion. In addition, she points out the importance of acknowledging suffering as part of the shared human experience. Thus, in our study, clients were affirming that it is normal to want to be loved and understood. Furthermore, they were approving that it is ok to be angry and frustrated. Likewise, Vráblová et al. [24] discovered in their qualitative study the categories ‘Universality of failure’ and ‘Understanding a situation’. Halamova et al. ([23] determined in their study of free associations of compassion and self-compassion a similar subdomain called understanding of the self. Additionally, our cases were showing self-compassion through gratefulness (I’m grateful for your support), appreciation (I appreciate you), and acceptance (I accept myself for who I am). According to the EFT, the theory promoting self-acceptance through the development of self-compassion is one of the key goals of the two-chair dialogue [11,59,71]. When individuals start being more self-compassionate and accept themselves for who they are, it increases their acceptance for others as well [74]. Thus, it is understandable that our clients were expressing gratefulness towards the critic and acknowledging the support. Furthermore, one client articulated missing closeness with the critic. In the case of this client, the critical voice was the adoptive mother. Based on the principal change process in EFT, grieving about the unmet needs and childhood losses is a substantial factor for emotional transformation. In addition, clients were giving themselves compassionate advice (I give you compassionate advice) by suggesting what to do and what not to do. As previously mentioned, the motivation to alleviate suffering is a key element of compassion [25,66]. According to Germer [68], giving kind advice (e.g., build up more self-esteem, take it easy, take care of yourself, accept that you are not perfect) and caring for one’s self are essential steps in growing self-compassion. Thus, suggesting to be more caring, not to do too many things, taking it easy, and having more awareness supports the research on the mental health benefits of self-compassion [34,75,76]. Similarly, Vráblová et al. [24] discovered that participants in their study were giving compassionate advice, such as ‘Be yourself’, ‘You can avoid a mistake’, and ‘You’re allowed to be imperfect’. Lastly, the compassionate voice proposed to be forgiving. In line with this, Zhang et al. [74] emphasise how self-compassion promotes self-acceptance, which improves acceptance for other imperfections and thus enhances forgiveness.

### 5.3. Emotional Aspect

#### 5.3.1. Self-Criticism

The emotional aspect is the least frequent domain for self-criticism. Similarly, Halamová et al. [20] identified the emotional aspect as well as the last domain participants associated with self-criticism. The self-critical voices in our study were criticising the lack of positive feelings, such as joy and happiness, being too depressed, and expressing unpleasant feelings, such as disgust, shame, anger, and contempt. As previously mentioned, self-criticism is linked to depression [57,59]. According to Greenberg and Watson [59] depressed clients criticize themselves for being depressed and not being happier. In alignment with this, it is generally acknowledged that self-criticism comes along with self-contempt [7,16], disgust [7], shame [62,77], and anger towards one’s self [7,70,78]. Our results are in line with recent consensual research by Halamova et al. [15]. Likewise, their participants were expressing contempt, disgust, and anger while being self-critical. Along the same lines, Whelton and Greenberg [7] acknowledge contempt as a common emotion that comes with self-criticism. Interestingly, in our analysed cases, clients were expressing more anger and disgust rather than contempt. We assume that this is due to the studio environment the sessions were recorded in. Though the analysed therapy sessions are real, the clients were not alone with the therapists. Consequently, clients were more cautious and expressed their criticism in a less contemptuous manner. Pascual-Leone et al. (2013) label “pervasive criticism coupled with a tone of contempt and disgust” ([70], p. 84) self-hate. As said by Kramer and Pascual-Leone [78], self-criticism is a form of maladaptive, rejecting anger directed towards one’s self, which covers more vulnerable feelings, such as shame. Hence, it is characteristic for clients to articulate a sense of unworthiness [79] as seen in our study. Hereby, our data deepens previous knowledge on emotions in self-criticism.

#### 5.3.2. Self-Protection

In terms of self-protection, the emotional aspect is the second most frequent domain. Clients were setting boundaries for critics saying that they do not want it to make them feel that way anymore. They were expressing anger and difficult feelings, such as hurt, disappointment and helplessness, towards the critic, and articulating pride. The subdomain ‘*I don*’*t want you to make me feel*’ was characterised by maladaptive emotions, such as fear and tiredness. Similarly, Vráblová et al. [24] found out that participants were verbalising the need to protect themselves from experiencing negative emotions. This mirrors the change process of the EFT model of emotional transformation [4,64]. Within the sequential model of emotional processing, maladaptive emotions (primary and secondary), such as fear, loneliness, and shame, are transformed into more helpful, primary adaptive emotions, such as protective anger and self-compassion [4,62]. Thus, protective anger (also known as assertive anger) evolves when clients come in contact with their unmet essential needs [4,62]. This facilitates self-acceptance and self-efficacy [4,5]. Consequently, they feel empowered to express positive emotions, such as pride. However, in our cases, only one client verbalised pride. Thus, clients were not fiercely articulating more positive emotions. As previously pointed out, our selected cases were one-time therapy sessions. Hence, clients were seeing the therapist for the first time in a recording studio. This supports the notion that emotion transformation is based on a strong therapeutic alliance and evolves over time [80,81]. When clients start feeling stronger and more resilient, they start standing up for themselves. Thus, in the subdomain ‘*I*’*m angry at you for…*’, clients were expressing their anger towards the critic for rejecting their feelings and living life through them. As stated by Timulak [11], expressing protective anger towards the critic allows clients to feel a sense of agency and strength simultaneously with self-compassion, which is an important factor to cope with self-criticism [4,11]. This is in line with Halamová et al. [82]. Their findings showed that individuals with a low level of self-criticism articulated anger towards their inner critics. Furthermore, our examples were articulating setting boundaries to experiencing difficult feelings, such as hurt, disappointment, and helplessness. The last two reflect the EFT term of global distress, which represents undifferentiated negative feelings ([4]. By verbalising their hurt, clients show their vulnerable feelings [11].

#### 5.3.3. Self-Compassion

In the matter of self-compassion, the emotional aspect was, with 14 statements, the least saturated domain in the study. Like self-protection, self-compassion is promoted in the third stage of the EFT change process [62]. Thus, analysing the first session of therapy decreases the possibility of clients transforming their self-criticism and expressing self-compassion as a primary adaptive emotion. In our study, clients were communicating empathy (I feel your pain) and positive feelings towards themselves (I have positive feelings for you). The categories ‘*You felt bad*’, ‘*It*’*s hard*’, and ‘*I*’*m sorry*’ are in accordance with Strauss et al.’s [25] definition of compassion. The third element identifies compassion as “feeling empathy for the person suffering and connecting with the distress (emotional resonance)” ([25], p. 19). Furthermore, there is a general agreement that compassion is partially also an emotion [83,84] characterised by warm and caring feelings [25,31,66] that arises when a person is confronted with another person’s suffering. According to Strauss et al. [25] self-compassion is compassion directed towards self. The self-compassionate reactions of our cases included positive feelings, such as pride and love. This goes along with the kindness aspect of Neff’s definition of compassion [31]. Conforming to this, participants in Pauly and McPherson’s study [67] perceived self-compassion as a concept that involves kindness. Interestingly, self-pride was expressed by the self-protective and self-compassionate voice. However, while the self-protective voice expressed pride for herself towards the critic, the compassionate voice articulated pride as a softer inner critic towards the experiential self. This supports the aim of the two-chair dialogue, which aims to soothe the inner critic to a compassionate voice and strengthen the experiential self through increasing self-protection [8,11].

### 5.4. Limitations

We analysed the verbal expression of self-compassion, self-criticism, and self-protection of clients in videos previously recorded in studio settings. Despite the videos being real therapy sessions and our aim to analyse natural therapy sessions, they were recorded in studios with cameras. Hence, we could not ensure that the setting was certainly natural. Additionally, clients were seeing the therapists for the first time surrounded by other people in the room. Consequently, they might have felt vulnerable and ashamed, which caused them to be more cautious in how they verbalise their statements. Furthermore, cameras can be intimidating and cause insecurities, which could have deformed the statements to be more appropriate. Another important limitation is the timing of the verbalisations of these states. According to Nardone et al. [85], there is a correlation between emotional arousal and the expression of unmet needs. As we eliminated the statements throughout the sessions, we could not examine the temporal influence on the verbalisation of self-compassion, self-protection, or self-criticism. Additionally, not all clients were expressing self-compassion, self-protection, and self-criticism. Hence, we had a different number of clients for each state. This goes along with the next condition, which was the inconsistency of our material. While six of our cases were single sessions, one client was analysed in two different and one in four different sessions. In accordance with this, given the natural conditions, the number of statements varied as we could not control how much the clients verbalised each state within the therapy sessions. Finally, it is noteworthy to refer to the low number of cases, to the absence of male clients, and to the concentration of the English language clients. Evidently, our findings cannot be generalised for men and other languages.

### 5.5. Implications for Practice

Our findings increase the knowledge on the productive articulation of these states. This provides practitioners with a verbal map that can help EFT therapists to guide clients to work through their self-criticism in the two-chair technique. Therapists could make use of our categories as a guide to empathically encourage clients to express self-compassion and self-protection. Similar to the emotion category system (ECCS; [53]) and the productivity model [44], our results can be used by practitioners as an inner compass that can support them to identify the effective verbal expression of each state. The results could equip clinicians with a directory of questions that can help clients to heighten self-criticism in an effective manner. For instance, therapists could ask clients to tell the experimental self what to do to keep up with the high standards or how to express high standards. Furthermore, clinicians can pay attention to the articulation of unpleasant significant feelings related to self-criticism, such as contempt, disgust, shame, or anger towards one’s self. Practitioners could encourage clients to verbalise their inner critical voice through the expression of these feelings. For example, therapists could ask clients to say what they are contemptuous about as the inner critic. In this way, therapists can lead clients to unfold their inner critical voice in an effective emotional state, which is a significant element within the EFT model of emotion transformation [1,5,8]. Having an inner road map of statements that promote articulating self-compassion is beneficial in moments when, in addition to following empathically, the therapist needs to lead the client towards a productive experience of self-compassion [44]. For example, therapists could ask clients to verbalise compassionate advice by suggesting what to do: ‘*What do you suggest her/him to do from a compassionate place?*’ or ‘*Can you see how much s/he is suffering? Could you understand her/his bad circumstances?*’. In this way, practitioners can employ our findings to guide clients to articulate self-compassion in a sufficient quality. In regard to self-protection, therapists can reinforce clients to tell the critic to see them for who they are or to stop negative behaviour or to set healthy boundaries. Furthermore, in order to promote a sense of empowerment, they can encourage clients to criticise the critic back or tell the critic why they do not have the right to be judgmental of them. Hence, our categories can be utilised by practitioners as a roadmap to establish an efficient articulation of self-protection that supports clients to fight for their needs and defend their self-worth (e.g., Timulak, 2015).

## 6. Conclusions 

Our study is the first qualitative research on the verbal expression of self-compassion, self-protection, and self-criticism in EFT. Over the last few years, Halamová and her team started to pay attention to the qualitative analysis of these states [15,20,23,24]. Our findings enhance the understanding of how clients articulate these states in therapy sessions in order to increase self-criticism and decrease self-compassion and self-protection in EFT two-chair work. So far, there is ample research acknowledging self-compassion as an antidote to self-criticism [26,68,86]. However, our findings demonstrate that both self-compassion and self-protection are key emotional processes in reducing the level of self-criticism. Future research could shed light on the timing of the verbalisation of these states as this could deepen the understanding of the order of verbalising self-compassion and self-protection in increasing self-criticism. Furthermore, it would be valuable to examine the clients’ transformation over a certain number of therapy sessions in order to enhance the knowledge on how the clients’ articulation of these states changes over time. Researchers could develop an observation guide as a measurement instrument for therapists to help them distinguish the three states easier and facilitate their occurrence. In addition, we propose future research to conduct an analysis in a more natural setting. Lastly, it would be beneficial to focus on a broader number of cases, gender differences, other languages, and cultures.

## Figures and Tables

**Table 1 ijerph-19-12942-t001:** Self-Criticism.

Domain	Subdomain	Category	Characteristic	Example
**Behavioural aspect**	I point out your wrong behaviours.	I put you down for past behaviours.	You always do…	“*You*’*re always arguing*”
			You sometimes do…	“*You sometimes forget to send lunch for your son…*”
			Your behaviours are not enough.	“*How hard you try it*’*s just not enough*
			You are not doing it right.	“*you*’*re not doing anything right*”
			You are not capable of…	“*you can*’*t even take your children to Disney World this year…*”
		I doubt your present behaviour.	What are you doing?	“…*now look at you what are you doing*?…”
		I mistrust your future actions	What are you going to do next?	“*what*’*s next, what do you gonna do next?*”
	I tell you what to do.	I tell you what to do instead.	Make more money.	“*with your college education, you should be making more money*”
			Obey me (self-critic).	“*you need to listen to what I tell you because…*”
			Push yourself.	“*Keep on doing it. Keep on pushing yourself.*”
			Work more.	“*you should go to school and take a class, you should get out…*”
		I tell you what not to do.	Don’t feel.	“*don*’*t feel just do*”
			Don’t cry.	“*…laugh and the world laugh with you, cry and you cry alone*”
			Don’t take time for yourself.	“*You don*’*t need to be taking time for yourself.*”
**Cognitive aspect**	I evaluate you as a person.	I have high standards towards you.	You are not meeting my standards.	“*you*’*re not living up to what I expect you*”
			I give you the conditions for being loved.	“*if you do what we say if you behave if you act the way we want you to then you are important to us*”
		You are not perfect.	You’re not the best.	“*okay you get straight As but you*’*re not in the top ten…*”
			You are good but.	“*you know you are a good wife but…*”
		You are inadequate.	You are not enough as a person.	“*you*’*re not good, you*’*re just not good…*”
			You don‘t have enough skills.	“*I mean you don*’*t even know how to…*”
			You are not efficient enough.	“*you know you can*’*t function*”
			You are not interesting enough.	“*You*’*re boring*”
		You are ridiculous.		“*you look you walk like you walking through cement…*”
	I evaluate your situation.	I criticize you for the situation you are in.	You are in a bad situation again.	“*…you fell right back into the same situation, and it*’*s the same thing*”
			It’s too hard for you again.	“*it*’*s too hard*”
			You are trapped again.	“*…you know what do you guys gonna do ahm…*”
			You are failing again.	“*this is your second marriage, and you can*’*t even make it work…*”
	You should be…	You should be more.	Be stronger.	“*you should be you should be been stronger*”
			Be more knowledgeable.	“*you should be known better that get involved with him in the first place…*”
			Have more self-esteem.	“*you should have more self-esteem*”
		You should be less.	Be less alone.	“*That*’*s alone, you should…*”
	You don’t deserve having needs.			“*there is no reason, there is no reason that you need to ever go out with your friends…*”
**Emotional aspect**	Your feelings are disproportional.	You feel too much of…	Depression.	“*yet lived depressed most of your life*”
		You feel too little of…	Happiness.	“*you not, you*’*re never happy*”
			Joy.	“*you can*’*t, can*’*t enjoy your children*”
	You are unpleasant to me.	Contempt.		“*Why are you even here?*”
		I’m disgusted by you.		“*I*’*m disgusted that you just lazy*”
		I’m ashamed of you.	You are unworthy.	“*you*’*re not worth anything*”
		Anger for…	letting it happen.	“*I*’*m angry that you that you let it happened*”
			being a victim.	“*I*’*m angry that you that you were a victim*”

**Table 2 ijerph-19-12942-t002:** Self-Compassion.

Domain	Subdomain	Category	Characteristic	Example
**Behavioural aspect**	I‘m going to alleviate your suffering.	I’m going to stop hurting you.	I will stop making demands.	“*I*’*m gonna stop making you these demand on you*”
			I will stop confusing you.	“*. I don*’*t wanna be confused anymore about the situation*”
			I will stop blaming you.	“*I*’*m gonna say you were you*’*re not to blame you were you were strong*
		I will show you support.	I‘m here for you.	“*I*’*ll be there when you need me and you can you can come to that place within yourself*”
			You did your best.	“*you did everything you could have done*”
			I give you safeness.	“*I*’*m not trying giving you answers*”
			I give you comfort.	“*I*’*m just trying to be the warmth that you*’*re seeking right know*”
			I give you something uplifting.	“*…I would call or take you somewhere or you know trying something that lift your spirit…*”
			It’s not my fault.	“*it wasn*’*t my fault*”
**Cognitive aspect**	I see your bad circumstances.	You were a victim.		“*and that she was a victim of the circumstance*”
		It was your childhood.		“*it*’*s what my situation what it gave me that*’*s that was my childhood*”
		It’s situational.		“*you know just a situational…*”
	You deserve all universal human needs.	It is normal to want to be loved.		“*I understand that you were trying to find somebody that loved you*”
		It is normal to want to be understood.		“*You need me to understanding you, so I try to understand you a little more.*”
	It’s ok to have difficult feelings.	It’s ok to be angry.		“*sometimes do I feel like I take out my Anger for towards my nan and you*”
		It’s ok to be frustrated.		“*…and my frustration about the situation I take that on you, I shouldn*’*t*”
	I missed closeness with you.			“*we could have grown together*”
	I’m grateful for your (to critic) support.	You taught me taking action.		“*even though you never gave me the words you did, you gave me the actions*”
		You helped me feel empowered.		“*…I know I got that, ahm sense of power per se*”
		You helped me feel strong.		“*you*’*ve given me strength and I don*’*t think I realized it till right now*”
	I appreciate you.	I appreciate myself for my positive qualities.		“*I do know that I can do anything I set my mind to, can do anything that I set my mind*”
		I appreciate myself for my achievements.		“…*I think about sitting there and think about all the things that I can do and I have done…*”
		You matter.		“*cause it*’*s not worth it, it*’*s not worth it , nobody is worth that*”
	I accept myself for who I am.	I’m ok.		“*I still came out I guess one say okay.*”
	I give you compassionate advices	I suggest you do.	Build more self-esteem.	“*If you find that self-esteem about yourself that I think that you need*”
			Take it easy.	“*definitely he would tell me take it easy*”
			Have faith.	“*you know you*’*re a child of God*”
			Take care of yourself.	“*…you*’*re gonna get sick again,*”
			Work it out.	“*try to work it out*”
			Show understanding towards…	“*we need to understand why…*”
			Be forgiving.	“*but we should also forgive*”
			Accept that you are not perfect.	“*that*’*s the best you can do*”
			Make a decision.	“*you have a decision to make*”
			Be more aware what’s going around so I can forgive you.	“*I can forgive you if you don*’*t put yourself into the same situation again*”
		I suggest what you shouldn’t do.	You do too many things.	“*you trying many things*”
			Don’t change.	“*you are not going to let it change you are not going to let it define who you are*”
			Don’t be dismissive.	“*don*’*t be so quick to dismiss him*”
**Emotional aspect**	I feel your pain.	You felt bad.		“*that you were feeling bad about yourself*”
		It’s hard.		“*…and that*’*s a hard way ever feel like*”
		I’m sorry for…	for treating you badly.	“*I don*’*t want you make you feel bad like this*”
			for shaming you.	“*And I don*’*t want to make you feel like you have to crawl up in a bowl and hide*”
	I have positive feelings for you	I love you.		“*I love you*”
		I’m proud of you.		“*I’m very proud of you*”

**Table 3 ijerph-19-12942-t003:** Self-Protection.

Domain	Subdomain	Category	Characteristic	Example
**Behavioural**	I tell you what I need.	I need you to see me for who I am.	I can’t be perfect.	“*I*’*m not going to be perfect and I*’*m gonna make mistakes*”
			I’m worthy.	“*I am valuable*”
			I have strength.	“*I was strong*”
			My past doesn’t define me anymore.	“*that what happened in the past doesn*’*t define me*”.
		I need you to stop negative behaviour to me.	Stop putting me down.	“*Stop putting me down*”
			Stop telling me what to do.	“*I don*’*t wanna hear you tell me what I should be doing*”
			Let go of the past.	“*I want you to let go of it as well*”
			Give me a break.	“*I need you to give me a break and get off my back*”
			Stop being rigid.	“*I don*’*t want to be I don*’*t want to follow structures and schedules*”
			Stop listening to others.	“*because it empowers me to kind of not internalize all those…*”
		I need you to do good to me.	Accept me.	“*I want to be accepted*”
			Be warm.	“*maybe I needed you to hug me and tell me that it was still okay instead of yelling at me*”
			Be supportive.	“*I need to be taken care of*”
			Be optimistic.	“*you also have to look at the positive things*” “*I want you to respect my children*”
			I need you to respect me.	“*I want you to respect my children*”
			Apologize for the damage you caused.	“*I would like you to acknowledge the damage that it cost*”
		I want to take responsibility for my life.	I want to be more expressive.	“*I want to be free to express myself and my feelings*”
			I want to be more flexible.	“*I want to live a slightly more unstructured life*”
			I want to take action.	“*I want to have fun, I want to be active and do things,*”
		I need to be good to myself.	Love myself	“*I need to love myself*”
			Take care of myself.	“*I need to take care of me and help me start changing my pattern…*”
			Take time for myself.	“*but I need to have some time for myself and I need you to understand that*”
		I need to set boundaries.	Stop it.	“*well I*’*m gonna say stop*”
			I won’t listen to you anymore.	“*I*’*m not doing the things that you would choose for me*”
			You don’t have the right.	“*that person doesn*’*t they don*’*t have the right to say those things to me*”
			I won’t accept being neglected anymore.	“*I can*’*t accept that you couldn*’*t think passed yourself and think about the bigger picture*”
			I reject your criticism.	“*I know it*’*s not true what it is that you*’*re telling me*”
**Cognitive aspect**	I show understanding towards myself.	I have a right to be myself.	I have my personal traits.	“*that*’*s doesn*’*t have to be exiting to be fun, to be okay*”
			I perform as I do.	“*I*’*ve done a good job, I*’*m doing a good job with my son*”
			I was on my stage of development.	“*I was on my stage of development, you can*’*t expect more from a child than what I did,*”
		I deserve to be loved.		“*I deserve to have mom who loved me*”
		You don’t have the right to judge me because…	It was matter of circumstances.	“*these things that just keep happening to me over and over.*”
			Don’t compare me because I’m different.	“*I don*’*t want you to ever say that I look like my nan again ever*”
			You don’t know me.	“*I can be a fun person to be around…*”
	I acknowledge your (the critic) protective function.	I acknowledge your effort.		“*you are getting better and I*’*m glad that you*’*ve taking these steps*”
		I acknowledge your reasons.		“*I do accept the reason I think I would…*”
	I criticize you back.	You don’t acknowledge me.		“*don*’*t you see anything good in me? if you do why don*’*t you ever say it?*”
		You’re forcing me.		“*then you tried to push me into things and I didn*’*t feel like I belong there and you just kept forcing me*”
		You don’t listen.		“*you wouldn*’*t listen to me when I tried to tell you this you just kept pushing me and pushing me*”
		You always change your standards.		“*you*’*re just like you*’*re nice and then you*’*re mean, and I can*’*t I can*’*t stay on that,*”
		You don’t care about me.		“*but I take care of you, completely and I get nothing…*”
		You’re not nice to me.		“*it*’*s not fair for anybody to treat you that way*”
**Emotional aspect**	I don’t want you to make me feel…	Frighten.		“I *don*’*t want to be frighten anymore*”
		Tired.		“*I*’*m tired of it*
	I’m proud of myself.			“*I*’*m proud of myself, for the accomplishments that I*’*ve made to this point,*”
	I’m angry at you for…	Rejecting my feelings.		“*I*’*m angry at you because I tried to tell you over and over again how I felt about things and you totally discounted my feelings*”
		Not being there for me.		“*…I*’*m angry at the fact that you drink, I know he has been drinking a lot lately, but you…*”
		Forcing me…		“*I am angry with you for forcing me into these situations*”
		Being so negative		“*…piss attitude, I*’*m angry with your attitude, it*’*s so nasty, so negative*”
		Being dismissive.		“*I*’*m angry that I wanna tried to talk to you about these things, you get defensive and tell me these things never happened*”
		For living your life through me.		“*I feel angry at you, because you tried to live your life through me*
	I’m hurt.			“*that hurts, I hurt…*”
	I’m disappointed.			“*then start being disappointed in me when I didn*’*t do as well as you expected me to do*”
	I feel helpless.			“*I didn*’*t know what to do and I carry a lot of that stuff around with me now and I don*’*t know what to do with it*”

## Data Availability

In order to comply with the ethics approvals of the study protocols, data cannot be made accessible through a public repository. However, data are available upon request for researchers who consent to adhering to the ethical regulations for confidential data.

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
