# Peer review of "Qualitative Analysis of Chair Tasks in Emotion-Focused Therapy Video Sessions"

_ijerph, 2022, doi:10.3390/ijerph191912942_

Round 1

Reviewer 1 Report

The study focus on the data lack of studies that empirically identify how clients articulate self-criticism, self-compassion, and self-protection in EFT-therapy two-chair interventions .

It is investigated how self-criticism, self-protection, and self-compassion are verbally expressed by clients within an EFT-therapy session during two-chair work?

Investigating the the concepts behavioral, emotional and cognitive aspects, it could be demonstratet that both self-compassion and self-protection are key emotional processes in reducing the level of self-criticism.

The study is limited by its qualitative analysis, it leaves out quantitative, i.e. empirically verifiable, connections or differences. However, the results are pioneering and pave a further research way and should therefore be published.

Implications for practice could contain even more concrete content-related recommendations for therapists that could be drawn from the results.

The summary should better relate the behavioral, emotional and cognitive levels to the variables self-criticism, self-protection, and self-compassion.

Author Response

Thank you for personally reviewing our manuscript and for giving us the opportunity to respond to your comments. We have made the relevant changes as detailed below and would like to resubmit our amended manuscript We hope you find our response detailed and satisfactory. If you have any further questions or comments, please do not hesitate to contact me.

Authors

REVIEWER 1

The study focus on the data lack of studies that empirically identify how clients articulate self-criticism, self-compassion, and self-protection in EFT-therapy two-chair interventions .

It is investigated how self-criticism, self-protection, and self-compassion are verbally expressed by clients within an EFT-therapy session during two-chair work?

Investigating the the concepts behavioral, emotional and cognitive aspects, it could be demonstratet that both self-compassion and self-protection are key emotional processes in reducing the level of self-criticism.

The study is limited by its qualitative analysis, it leaves out quantitative, i.e. empirically verifiable, connections or differences. However, the results are pioneering and pave a further research way and should therefore be published.

Implications for practice could contain even more concrete content-related recommendations for therapists that could be drawn from the results.

This is a very valuable feedback. Thank you. We explained furher implications more in depth with clear suggestion for practicioners.

The summary should better relate the behavioral, emotional and cognitive levels to the variables self-criticism, self-protection, and self-compassion.

Yes, we could have also differentiated the discussion the way you suggest. We will keep it my mind for further research. Thank you.

Submission Date

10 September 2022

Date of this review

16 Sep 2022 23:26:27

Reviewer 2 Report

Thank you very much for offering me the oppotunity to review this paper. The paper focused on analyzing qualitatively the articulation of self-criticism, self-compassion and self-protection in the context of EFT. The results of the paper are very interesting and are adding something new to the literature. In general the paper was well-written and provided the necessary details in the Introduction, Result and Discussion sections. I would suggest that this paper is accepted after minor revisions. The main corrections would be: 1)  to add more about the potential implications in the introduction to strengthen the importance of the paper from the beginning, which could be linked later to the discussion and  2) more information on the development of domains, subdomains and categories in the "Materials and Methods" section. Some further revisions are suggested below.

Introduction:

1) The authors are clearly presenting the processes underlying EFT, the concepts of interest and the research behind them. However, the introduction would have benefited from subheadings to make it clearer to the reader.

2) In line 71, the authors mention “... and the kind of statements clients express in therapy sessions”. This sentence would have benefited from some examples to make it clearer to the reader what the authors have in mind, since every reader has different examples in mind.

3) The paper is missing of a strong argument of why it is important to study the articulation of self-criticism, self-compassion and self-protection in EFT-therapy sessions. So far the argument coming across to the readers is that these concepts are articulated in EFT-therapy and there is not enough research on them. But what would be the clinical implications if we had this kind of data? The authors refer to the clinical implications in the discussion section but it would be nice to have a brief introduction to them early in the paper.

Materials and Methods:
1) The paper could benefit from a diagram demonstrating the review process, e.g. how the 17 EFT sessions were reduced to 12. How many were excluded because of the language or because they didn’t mention any of the concepts of interest? If most of the sessions excluded are not mentioning the concepts of interest – does this show something for their usage in EFT?

2) Could the authors provide more information on how they developed the different domains, subdomains and categories?

Results:

1) The information provided in lines 220 until 228 and the sentence starting “For self-protection, all data...” starting in line 230 and finishing in line 231 should be moved to the “Materials and Methods” section. The Result section should only include the results and their qualitative interpretation.

2) Table 1 , p. 7 next to “Emotional aspect” – there is an extra space in the word “feelings”. Please delete!

3) Overall nice and structured qualitative analysis section.

Discussion:

1)   Overall very nice summary of the data and connection to previous studies. The implication sections was very short but to the point.

2)   There is a spelling mistake in Line 426 - the word “add” is written as “ad”. Please revise. 

Author Response

Thank you for personally reviewing our manuscript and for giving us the opportunity to respond to your comments. We have made the relevant changes as detailed below and would like to resubmit our amended manuscript We hope you find our response detailed and satisfactory. If you have any further questions or comments, please do not hesitate to contact us.

Authors

REVIEWER 2

Thank you very much for offering me the oppotunity to review this paper. The paper focused on analyzing qualitatively the articulation of self-criticism, self-compassion and self-protection in the context of EFT. The results of the paper are very interesting and are adding something new to the literature. In general the paper was well-written and provided the necessary details in the Introduction, Result and Discussion sections. I would suggest that this paper is accepted after minor revisions. The main corrections would be: 1)  to add more about the potential implications in the introduction to strengthen the importance of the paper from the beginning, which could be linked later to the discussion and  2) more information on the development of domains, subdomains and categories in the "Materials and Methods" section. Some further revisions are suggested below.

Thanks a lot. Both suggesstions are added to each section.

Introduction:

1) The authors are clearly presenting the processes underlying EFT, the concepts of interest and the research behind them. However, the introduction would have benefited from subheadings to make it clearer to the reader.

Thanks, we added more subheadings to the introduction as well as numbering the headings and subheadings in the whole manuscript.

2) In line 71, the authors mention “... and the kind of statements clients express in therapy sessions”. This sentence would have benefited from some examples to make it clearer to the reader what the authors have in mind, since every reader has different examples in mind.

Thank you for this suggestion. We added statement to it.

3) The paper is missing of a strong argument of why it is important to study the articulation of self-criticism, self-compassion and self-protection in EFT-therapy sessions. So far the argument coming across to the readers is that these concepts are articulated in EFT-therapy and there is not enough research on them. But what would be the clinical implications if we had this kind of data? The authors refer to the clinical implications in the discussion section but it would be nice to have a brief introduction to them early in the paper.

We added further explanation in regards to implications for practice to the section The aim of the research study. More implications for practicioners are mentioned under Implications.

Materials and Methods:
1) The paper could benefit from a diagram demonstrating the review process, e.g. how the 17 EFT sessions were reduced to 12. How many were excluded because of the language or because they didn’t mention any of the concepts of interest? If most of the sessions excluded are not mentioning the concepts of interest – does this show something for their usage in EFT?

None of the recordings were excluded because of the language. Our main goal was identiying EFT-sessions in which the working phase included the EFT two-chair or empty-chair task. Unfortunately, not all recordings involved these tasks in a productive way. Furthermore, we identified the expression of self-criticism, self-protection, and self-compassion according to the EFT-definition of these states. The states had to be expressed in a productive manner in order to be useful for our study. We added the criteria we used for the selection of each state to the procedure section.

2) Could the authors provide more information on how they developed the different domains, subdomains and categories?

Thank you for your interest in the CQR procedure. We explained the procedure furthermore in the paper.

Results:

1) The information provided in lines 220 until 228 and the sentence starting “For self-protection, all data...” starting in line 230 and finishing in line 231 should be moved to the “Materials and Methods” section. The Result section should only include the results and their qualitative interpretation.

Thank you for your feedback. We deleted this section and added it to the part procedure and data analysis.

2) Table 1 , p. 7 next to “Emotional aspect” – there is an extra space in the word “feelings”. Please delete!

Done.

3) Overall nice and structured qualitative analysis section.

Thank you!

Discussion:

1)   Overall very nice summary of the data and connection to previous studies. The implication sections was very short but to the point.

Thank you for your nice feedback. We added some more suggestions for implications to the section.

2)   There is a spelling mistake in Line 426 - the word “add” is written as “ad”. Please revise. 

Done!

Submission Date

10 September 2022

Date of this review

28 Sep 2022 14:30:03